# A Comprehensive Review of Land Use and Land Cover Change Based on Knowledge Graph and Bibliometric Analyses

Caixia Rong [1,2,3,4] and Wenxue Fu [1,4,*]

1 International Research Center of Big Data for Sustainable Development Goals, Beijing 100094, China; rcx@mail.bnu.edu.cn
2 Institute of Remote Sensing Science and Engineering, Beijing Normal University, Beijing 100875, China
3 State Key Laboratory of Remote Sensing Science, Beijing 100101, China
4 Key Laboratory of Digital Earth Science, Aerospace Information Research Institute, Chinese Academy of Sciences, Beijing 100094, China
* Correspondence: fuwx@radi.ac.cn

**Abstract:** Land use and land cover (LULC) changes are of vital significance in fields such as environmental impact assessment and natural disaster monitoring. This study, through an analysis of 1432 papers over the past decade employing quantitative, qualitative, bibliometric analysis, and knowledge graph techniques, aims to assess the evolution and current landscape of deep learning (DL) in LULC. The focus areas are: (1) trend analysis of the number and annual citations of published articles, (2) identification of leading institutions, countries/regions, and publication sources, (3) exploration of scientific collaborations among major institutions and countries/regions, and (4) examination of key research themes and their development trends. From 2013 to 2023 there was a substantial surge in the application of DL in LULC, with China standing out as the principal contributor. Notably, international cooperation, particularly between China and the USA, saw a significant increase. Furthermore, the study elucidates the challenges concerning sample data and models in the application of DL to LULC, providing insights that could guide future research directions to accelerate progress in this domain.

**Keywords:** LULC; DL; bibliometrics; knowledge graph

## 1. Introduction

Land use and land cover (LULC) embody the essential traits of the Earth's system, significantly intertwined with numerous human endeavors and the physical surroundings [1]. LULC intimately intersects with human livelihood and productive activities, serving as a pivotal component in land utilization and global environmental shifts [2]. Insights into LULC alterations are indispensable across a plethora of fields leveraging Earth observations, including urban and regional orchestration [3,4], gauging environmental susceptibility and implications [5–7], shifts in climate [8–10], surveillance of natural calamities and threats [11,12], alongside appraisals of soil attrition and salinization among others [13,14]. In the span of the preceding decades, we have witnessed an exponential surge in satellite or airborne spatial imagery and data, largely attributable to the emergence of remote sensing technologies and the deployment of an array of satellites [15]. Remote sensing technology has provided new technical means for LULC due to its advantages of offering wide coverage and accessing large amounts of information [16]. Remote sensing images have also become the main data source in the study and monitoring of LULC changes [17].

Since 2012, when the neural network model AlexNet, developed by Hinton et al., achieved remarkable results in the ImageNet image recognition competition, the field of deep learning (DL) has undergone robust development [18]. DL was named one of the top ten breakthrough technologies of 2013 [19]. In the following years, DL achieved significant



advancements in areas such as image recognition, speech recognition, and natural language processing. The rapid development of DL technology has provided new opportunities and challenges in its application in LULC classification [20]. Consequently, various models utilizing machine learning (ML) and DL for the analysis of LULC have been developed. These models can be categorized into three types based on their input data: pixel-based methods, spatial methods, and sequence methods [21]. Traditional pixel-based methods classify each pixel individually based on their corresponding spectral data, such as random forests (RF) [22,23], support vector machines (SVM) [24], and self-organizing maps (SOM) [25,26]. Chaitanya B. Pande et al. [27] introduced a RF learning algorithm based on the Google Earth Engine (GEE) platform, utilized for creating maps of LULC in India and conducting change detection mapping through the SAGA GIS software. Riese, F.M. et al. [26] proposed the SOM framework for unsupervised land cover type classification of hyperspectral data. Spatial classification methods classify not only using a single pixel but also a two-dimensional (2D) spatial neighborhood. The typical approach is based on 2D convolutional neural networks (CNNs), composed of filter layers that perform hierarchical learning; learning low-level features in the first layer, and higher-level features in the last layer. Zhang et al. [28] employed a joint DL (JDL) model of a multilayer perceptron within CNNs to simulate LULC classifications and change at two sites in Southampton and Manchester, UK. He et al. [29] developed an integrated model of CNN with a cellular automaton (CA) to simulate urban development in the Pearl River Delta of China. Sequence methods include recurrent neural networks (RNN), long short-term memory (LSTM) networks, and 3DCNNs. Chen et al. [30] proposed a deep Siamese convolutional multi-layer RNN for change detection in sequential high-resolution images. Geng et al. [31] proposed the ST-CA, which employs a potential generation module using a 3DCNN to calculate the development potential of each cell, and a spatial allocation module using a patch-based CA to simulate future LUCC.

Prior to the emergence of DL technology, the traditional methods for LULC modeling and assessment using remote sensing images primarily relied on handcrafted features and machine learning algorithms [32]. Manual design of features demands high domain expertise and requires significant human and material resources [33]. In contrast to machine learning, DL does not require manual feature design. It offers an end-to-end approach to LULC modeling, is capable of automatically extracting features from data, and exhibits stronger robustness and generalizability [34]. Given the numerous advantages of DL and remote sensing technologies, DL has become a focus of considerable attention in the study of LULC [35]. Therefore, over the past few years, there have been many reviews and discussions about ML and DL in the field of remote sensing [36–38]. However, existing reviews mainly focus on the image processing and segmentation recognition techniques of DL, and are mainly applied to the analysis of remote sensing images. These studies have not objectively and systematically analyzed the significant issues, development trends, and existing problems in this field.

Bibliometrics is a method of analyzing and evaluating the literature using quantitative methods, providing insights into the quantity, quality, influence, development trends, and relationships among scholarly publications [39]. This discipline capitalizes on various tools such as HistCite [40], SATI [41], CiteSpace [42], which are adept at importing and transmuting data from diverse bibliographic databases, including WoS (WoS), Scopus, Dimensions, and Lens [43]. These tools furnish researchers with a comprehensive array of capabilities for literature information analysis and visual representation of results. Bibliometrics has been extensively applied across various fields. Pacheco Quevedo R et al. [44] conducted a bibliometric analysis and review on how LULC is explored in the context of landslide susceptibility in 536 scientific articles spanning from 2001 to 2020. Pham-Duc B et al. [45] employed a bibliometric methodology to analyze articles related to GEE within the Scopus database. However, no study has yet applied bibliometrics to examine the application of DL models in LULC. Knowledge graphs are analytical research tools that present large amounts of information from the literature in graphic form, revealing re-

search hotspots and development trends [46]. Using knowledge graphs and bibliometric analysis can assist in objectively and systematically analyzing the research progress of and development trends in DL algorithms in LULC applications [44].

In this study, our aim was to ascertain the advancements of DL models in LULC research by employing knowledge graphs and bibliometrics to systematically analyze and summarize the literature in this field, both qualitatively and quantitatively. Finally, we discuss the primary research topics, the challenges present in applying DL to the LULC domain, and current advancements concerning models and sample datasets, thereby providing a reference for researchers conducting related studies. This research is guided by the following questions:

RQ1: what are the trends of articles and citations in this field?

RQ2: what are the most prolific sources of publications, countries/regions, and institutions?

RQ3: what is the nature of scientific collaborations between key countries/regions and institutions?

RQ4: what are the primary research topics of interest within this field?

RQ5: what are the topic distributions across major sources of publications, subjects, countries/regions, and institutions?

RQ6: what are the main advancements in models and sample datasets within this field?

## 2. Data Sources and Methods

### 2.1. Data Source

WoS is a comprehensive interdisciplinary academic information resource, managed by Clarivate Analytics, which includes a multitude of core academic journals and publications [47]. This database provides advanced citation analysis and metrics to help researchers evaluate and track the impacts of and trends in their research output [48]. We utilized the WoS Core Collection as our data source and adopted the topic search (TS) method for retrieval. The following three topics were searched in the literature: 'Land Use Change/Cover', 'DL', and 'Remote Sensing'. In an attempt to retrieve as many relevant articles as possible, we tried different combinations of search terms, using the Boolean operators AND/OR for combination searches. The search type was set as '(TS = DL) and TS = (land use and land cover) and TS = (Remote Sensing)'. The literature search spanned from 2013 to 2023, with the WoS journal database serving as the data source for the past decade. A total of 1432 articles were retrieved. After excluding books, book chapters, conference proceedings, reports, as well as 'grey literature', theses, and dissertations [49], we were left with 1310 articles. These articles were used as the sample data for our research analysis.

### 2.2. Methods

For this study, we mainly adopted two research methods: bibliometric analysis and knowledge graph analysis. In this research, a methodological framework, consisting of three stages: data retrieval, data cleaning, and data analysis, was developed, as shown in Figure 1. Various bibliometric indicators were applied to evaluate the included publication sources, disciplines, institutions, and countries/regions in the study. These indicators include: the number of articles, citation counts, the h-index, and the average citations per paper (ACP). The bibliometric analysis was principally divided into three parts: (i) assessing the productivity and impact based on the number of publications and citations; (ii) utilizing knowledge maps to intuitively display the collaboration relationships between authors and countries; (iii) identifying common keywords and research fields to highlight the main research themes. In the first part of the analysis, trends in scientific output were examined, primarily through considering the publications, citations, references in each paper, and the focus on highly-cited papers. Subsequently, network graphs generated by VOSviewer software were used to analyze co-authorship relationships [50]. Furthermore, a mapping of publication counts by country and research field was conducted to reveal the geographical distribution of the research and to determine which countries and research scales are most

frequently studied. Frequently appearing keywords facilitated a deeper understanding of technical terms, structural consistency [51], and trends in thematic areas [52]. Tools such as R language and CiteSpace software were used for the analysis of abstract keywords. Lastly, the most commonly used models in the LULC research field, as well as the issues and progress in the sample data, were discussed. The aim was to provide a comprehensive overview and to offer insights for future research directions in this field.

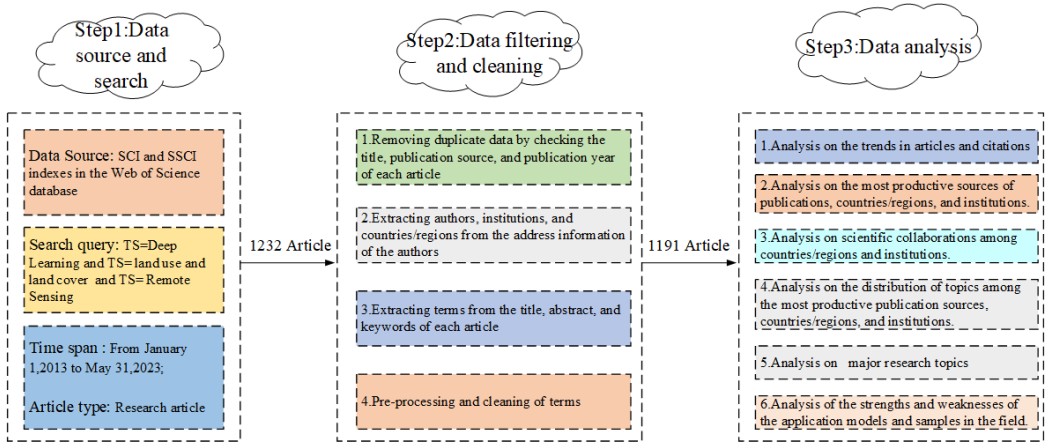

**Figure 1.** Research design and workflow.

## 3. Results of Bibliography Analysis

### 3.1. Number of Published Articles and Citation Trends

This study provides a visual analysis of the annual trends in the numbers of articles and citations in the field of LULC research, which utilizes DL model technology, as depicted in Figure 2. Our work showed that the research interest in this field has been gradually growing, as was particularly evident when the number of research findings reached 450 articles in 2022. The number of citations for articles in 2019 reached 5335, and in 2020, this number increased to 5499. The decreases in citation counts in 2021 and 2022, along with a drop in the number of research findings in 2023, were due to the time required for publications to be indexed in the database, and we could not include all of the articles published in 2023. However, looking at the significant growth trend in the number of articles and citations from 2013 to 2020, we can determine that since the development of DL technology, its application in LULC research has received considerable attention in academia. Consequently, the future research development prospects in this field are broad, and the number of papers about it is expected to continue to grow.

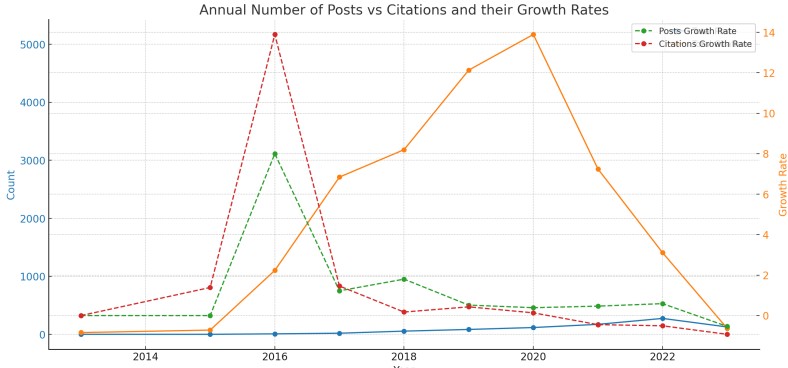

**Figure 2.** Annual Growth Trend Chart of LULC from 2012 to 2023. The solid lines represent the number of articles each year (blue) and the number of citations (orange), while the dashed lines represent the annual growth rate of articles (green) and the growth rate of citations (red). The left *y*-axis represents the count, while the right *y*-axis represents the growth rate.

### 3.2. Top Publication Sources

The 1310 articles retrieved in this study came from 258 publications, with the top five journals having published at least 20 related articles each. *Remote Sensing* was the journal with the most articles in this field, with 269 articles, followed by *IEEE Transactions on Geoscience and Remote Sensing* (91 articles), the *IEEE Journal of Selected Topics in Applied Earth Observations and Remote Sensing* (87 articles), and the *International Geoscience and Remote Sensing Symposium* (*IGARSS*; 64 articles). The journals with the top three highest numbers of citations were *Remote Sensing*, the *ISPRS Journal of Photogrammetry and Remote Sensing*, and *IEEE Transactions on Geoscience and Remote Sensing*.

According to the results in Table 1, among the top five publications with the most articles published, the top three were *Remote Sensing* (with an H-index value of 38), *IEEE Transactions on Geoscience and Remote Sensing* (26), and the *ISPRS Journal of Photogrammetry and Remote Sensing* (26). In terms of the ACP for each journal, the top three were *Remote Sensing of Environment* (with an ACP value of 78.47), the *ISPRS Journal of Photogrammetry and Remote Sensing* (76.55), and *IEEE Geoscience and Remote Sensing Letters* (47.97). Taking into account the aforementioned common indicators for evaluating papers, the work published in the *ISPRS Journal of Photogrammetry and Remote Sensing*, *IEEE Transactions on Geoscience and Remote Sensing*, and *Remote Sensing of Environment* deserves attention. As high-ranking journals in the field of geography, these publications have had significant impacts on such geoscientific research as remote sensing and geoinformation, and have played crucial roles in promoting the development of this field.

**Table 1.** Publication sources.

| Publication Sources | A | C | H | ACP | IF (Q) |
|---|---|---|---|---|---|
| *Remote Sensing* | 269 | 5260 | 38 | 20.56 | 5.601 (Q2) |
| *IEEE Transactions on Geoscience and Remote Sensing* | 91 | 3018 | 26 | 33.95 | 8.125 (Q1) |
| *IEEE Journal of Selected Topics in Applied Earth Observations and Remote Sensing* | 87 | 1875 | 22 | 21.93 | 4.715 (Q3) |
| *International Geoscience and Remote Sensing Symposium, IGARSS* | 64 | 579 | 10 | 9.17 | 7.49 (Q2) |
| *ISPRS Journal of Photogrammetry and Remote Sensing* | 49 | 3705 | 26 | 76.55 | 11.774 (Q1) |

Note: A: article count; C: citation count; H: H-index; ACP: average citations per paper; IF (Q): impact factor in 2023 and JCR ranking.

### 3.3. Highly Productive Institutions and Countries/Regions

The 1310 collected papers on the application of DL in LULC were contributed by 92 countries/regions (Figure 3), with a minimum of 36 articles published by the top five countries. China has published the most papers in this field, with a total of 607 articles, followed by the United States (193 articles) and India (135 articles). According to the H-index, the top three contributing nations were China (H-index of 52), the United States (H-index of 36.51), and Germany (H-index of 25) (Table 2).

**Table 2.** Top countries/regions.

| C/R | A | C | H | ACP |
|---|---|---|---|---|
| China | 607 | 11,730 | 52 | 20.83 |
| USA | 193 | 6865 | 36.51 | 38 |
| India | 135 | 658 | 15 | 5.03 |
| Germany | 97 | 3095 | 25 | 32.67 |
| France | 69 | 2311 | 19 | 33.96 |

Note: C/R: country/region; A: article count; C: citation count; H: H-index; ACP: average citations.

According to the ACP, the top three contributing nations are the USA (with an ACP value of 38), France (ACP value of 33.96), and Germany (ACP value of 32.67). Even though France and Germany had fewer articles (fourteen and seven, respectively) compared

to the leading countries/regions, their article quality was higher, as evidenced by the ACP analysis.

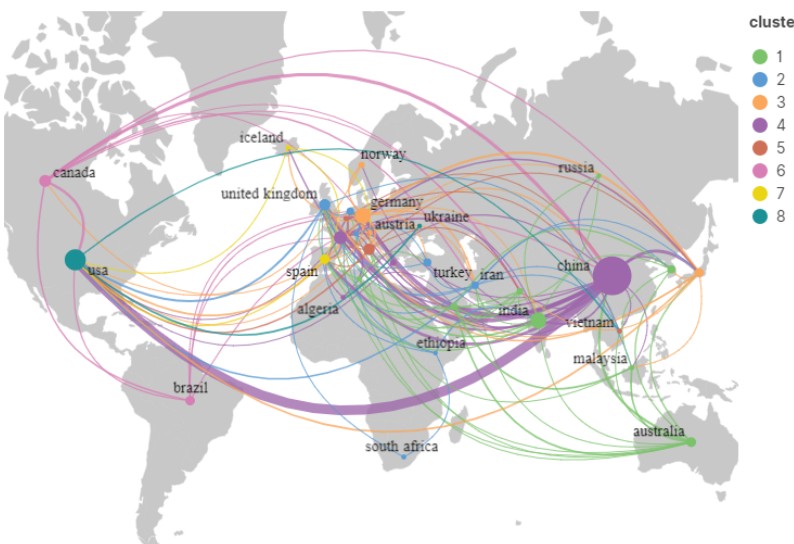

**Figure 3.** Countries of origin for published articles.

The 1310 articles retrieved were contributed by 198 institutions, each of which contributed at least four articles in this field. The top five institutions that published articles in this field are listed in Table 3.

**Table 3.** Top institutions.

| Institutions | C/R | A | C | H | ACP |
|---|---|---|---|---|---|
| Chinese Academy of Sciences | China | 119 | 2603 | 24 | 22.4 |
| Wuhan University | China | 87 | 3226 | 27 | 37.93 |
| Helmholtz Association | Germany | 59 | 1326 | 17 | 23.22 |
| German Aerospace Centre DLR | Germany | 34 | 1923 | 15 | 57.06 |
| Centre National de la Recherche Scientifique CNRS | France | 27 | 1046 | 13 | 39.15 |
| Udice French Research Universities | France | 25 | 1168 | 13 | 47 |
| Technical University of Munich | Germany | 21 | 1718 | 12 | 82.38 |
| Nanjing University | China | 20 | 311 | 10 | 16.1 |
| Beijing Normal University | China | 18 | 226 | 8 | 12.67 |
| Xidian University | China | 18 | 638 | 10 | 36 |

Note: C/R: country and region; A: article count; C: citation count; H: H-index; ACP: average citations per article.

Among the top ten institutions listed, five are from China, indicating the active role of China in the field of LULC research using DL. The Chinese Academy of Sciences is the institution with the highest number of articles in this field, having published 119 articles, followed by Wuhan University (87 articles) and the Helmholtz Association (59 articles). According to the citation count analysis, the top three institutions are Wuhan University (3226 citations), the Chinese Academy of Sciences (2603 citations), and the German Aerospace Center DLR (1923 citations). From the H-index analysis, the top three institutions among those listed are Wuhan University (with an H-index of 27), the Chinese Academy of Sciences (H-index of 24), and the Helmholtz Association (H-index of 17). From the ACP index analysis, the top three institutions are the Technical University of Munich (with an ACP value of 82.38), the German Aerospace Center DLR (ACP value of 57.06), and UDICE French Research Universities (ACP value of 47).

When all of the aforementioned indicators are taken into account, the performances of Wuhan University and the University of Chinese Academy of Sciences are commendable, demonstrating their significant contributions to the research in this field.

### 3.4. Scientific Collaboration

In Figure 4, organizations and countries/regions are represented as nodes and their collaborations are represented as lines that make up edges, with the intensities of the collaborations reflected in the thicknesses of the lines. The size of each node is proportional to the output of the organization or country/region it represents, and the width of each line represents the number of collaborative papers between organizations or countries/regions. Furthermore, each color represents the continent of a organizations country/region or that of an organization. Among the 1310 articles retrieved, 33 countries contributed more than three collaborative articles each. Among these thirty-two countries, fifteen are from Europe, nine are from Asia, two are from North America, one is from South America, one is from Oceania, and four are from Africa. The United States, China, the Netherlands, Germany, the United Kingdom, and France have collaborated with 24, 17, 16, 15, 15, and 15 other countries/regions, respectively.

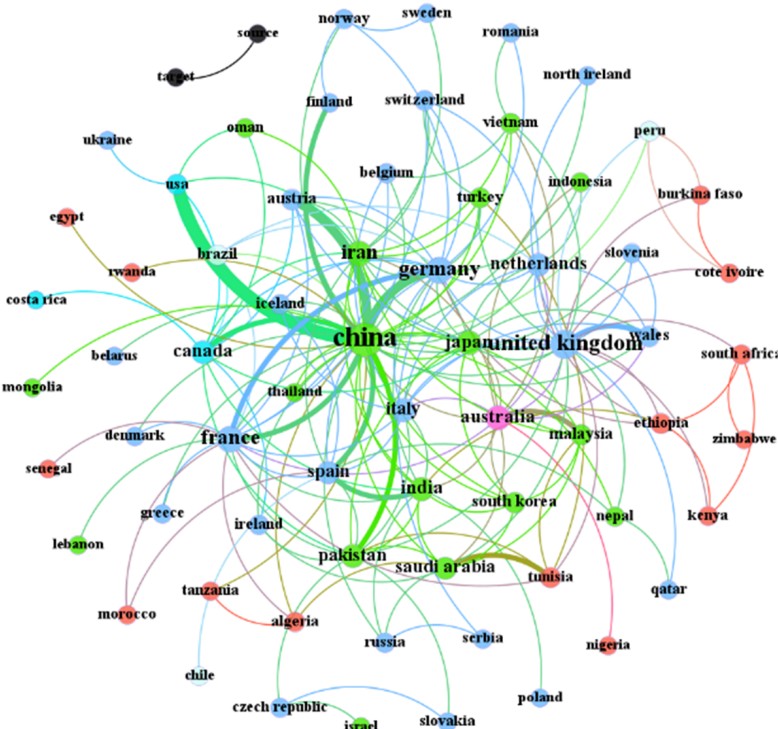

**Figure 4.** Country/region cooperation visualization graph.

According to the knowledge graph above, the most frequent collaborations were between the United States and China, with 21 cooperative articles. The combination with the next most frequent collaborations is Australia and China (eight), followed by China and Germany (seven).

Figure 5 illustrates the scientific collaborations among the top 50 research institutions in the field, based on the number of papers published. Among these fifty institutions, thirty-two are from China, three are from Germany, two are from the Netherlands, and two are from the United States. The Chinese Academy of Sciences, Wuhan University, and Sun Yat-sen University were the most collaborative institutions, partnering with 21, 15, and 12 other institutions, respectively. The University of the Chinese Academy of Sciences and the Chinese Academy of Sciences collaborated on nineteen papers, followed by Wuhan University and the University of the Chinese Academy of Sciences (ten papers), and Wuhan University and Sun Yat-sen University (seven papers).

In this field, the collaborations between research institutions are predominantly centralized around the Chinese Academy of Sciences and Wuhan University, which demonstrated the highest degree of collaboration. Sun Yat-sen University and the German Aerospace Center are also highly central in the collaboration network. These institutions, which publish frequently and occupy central positions, have always been at the forefront of research on land use and cover dynamics (LUCDs) using DL.

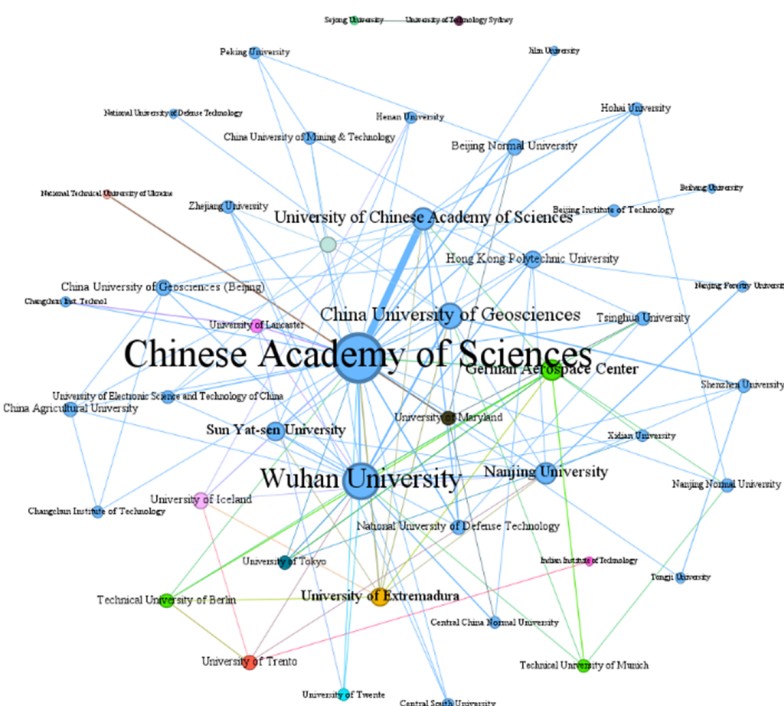

**Figure 5.** Research institutions collaboration map.

### 3.5. Networking Analysis Using the Key Research Terms

Keywords are the core of academic papers; they are high-level summaries of the content [53]. Applying the Structural Topic Model (STM) [54] to the abstracts and titles of the papers, we extracted ten significant research themes, as depicted in Figure 6. The analysis revealed that 'Urban Studies' emerged as the most extensively investigated theme in the realm of remote sensing, followed by 'Forest and Land Studies', 'Change Detection and Cloud Imaging', 'Land Cover Classification', 'Network and Feature Attention', 'Sentinel Time Series', 'Neural Image Classification', 'SAR and Land Cover Mapping', 'High Resolution Semantic Segmentation', and 'Crop and Domain Data'. Each sector within the rose diagrams corresponds to the frequency of the respective theme within a particular year, providing a lucid depiction of the yearly research focus.

The rose diagrams illustrate the annual distribution of the detected themes, indicating that the utilization of DL models in remote sensing research has become increasingly diverse over time (Figure 6). This reflects a wide-ranging spectrum of issues attracting scholarly attention. For instance, in 2012, the research community exhibited a notable inclination towards 'Urban Studies'. During 2013 and 2015, 'Forest and Land Studies' primarily dominated the research focus. The year 2016 marked a significant upsurge in interest in 'Change Detection and Cloud Imaging'. Since 2017, researchers in the field of remote sensing have demonstrated growing interest in 'Land Cover Classification', 'Network and Feature Attention', and 'Sentinel Time Series'.

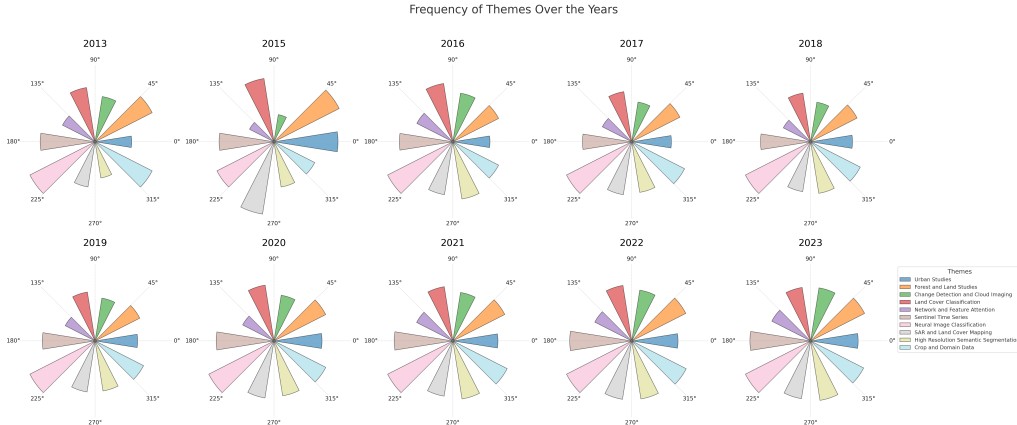

**Figure 6.** Rose diagrams depicting the distribution of research themes. Each sector represents a theme, its length indicates the number of papers associated with that theme, and its color distinguishes between different themes.

Figure 7 displays the distribution of research themes across prolific countries and institutions. Each of the top countries and institutions showcases a broad range of research interests in remote sensing, indicating their unique research strengths and focus areas. For instance, the People's Republic of China shows a strong inclination towards research in the theme of 'Urban Studies' and 'Forest and Land Studies', indicating its significant contributions in these areas. Germany, on the other hand, has displayed considerable involvement in the 'Neural Image Classification' and 'High Resolution Semantic Segmentation' themes. Institutions also reflect specific interests in certain research themes. The Indian Institute of Technology (IIT) System, for instance, has displayed a significant focus on 'Urban Studies' and 'Change Detection and Cloud Imaging'. The National Institute of Technology (NIT) System is heavily involved in the 'Forest and Land Studies' and 'Land Cover Classification' themes. Moreover, China University of Geosciences shows a significant contribution to 'Network and Feature Attention' and 'Sentinel Time Series'.

A diverse range of research themes can also be observed in other top institutions, such as the Chinese Academy of Sciences and Wuhan University. This highlights their wide-ranging contributions to different facets of remote sensing research.

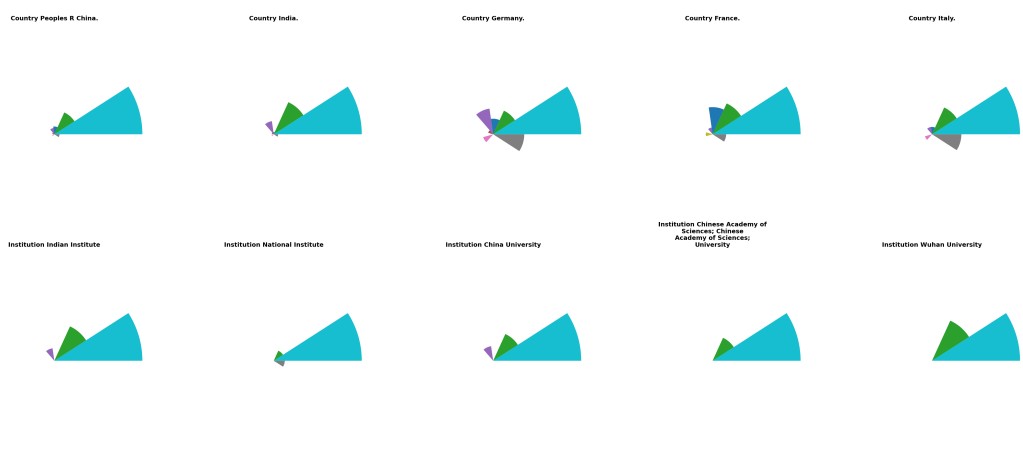

**Figure 7.** Rose diagrams depicting the distribution of research themes across the top five countries in research. Each sector represents a theme, its length indicates the number of papers associated with that theme, and its color distinguishes between different themes.

## 4. Discussion

This study employs econometric methods to examine the current status, trends, and key issues of LULC research using DL models. The overall increase in scientific articles and their citation counts reflects a significant growth in interest and impact in this research field within the study period. Analyses based on publication sources, disciplines, institutions, and countries/regions demonstrate that articles contributing to the innovation and application of LULC models have been recognized and become popular, particularly during 2018–2022. From the perspective of high-yield institutions and countries/regions, China has participated in nearly 50% of the research, with five of the top ten high-yield institutions originating from China. Results from the scientific collaboration analysis suggest a close cooperation among institutions and countries/regions, but it is recommended that inter-regional and inter-institutional cooperation should be enhanced. Despite significant advancements in this field, certain issues persist. This paper discusses these issues, specifically those related to sample datasets and models.

### 4.1. Data Sample

Despite significant advancements in the use of DL for image classification in LULC over the past few years, several challenges have persisted [55]. First, DL requires a substantial volume of training data, leading to high costs in data acquisition and annotation. Second, types and characteristics of land cover vary across different regions, necessitating region-specific data for model training [56]. Furthermore, the effective integration of multi-source data (optical imagery, radar data, elevation data, etc.) is an important research direction.

In recent years, scholars from various countries have released a series of sample LULC classification datasets [57–67] that cover different scales, sensor types, time intervals, spatial resolutions, and spectral resolutions. These provide fundamental data for related research in this field. Existing publicly available datasets in the LULC field can be divided into pixel-level and object-level samples. Pixel-level samples come from semantic segmentation datasets, with different LULC boundaries annotated with the pixel unit; object-level samples come from scene recognition datasets, with annotation performed using one type of LULC as the unit.

#### 4.1.1. Pixel-Level LULC Remote Sensing Classification Dataset

The aim of pixel-level classification techniques is to assign a category label to each pixel in an image. As a form of pixel-level classification, semantic segmentation annotates the category of each pixel based on context information. In recent years, fully convolutional networks (FCNs) [68] have attracted wide attention due to their outstanding performance in semantic segmentation tasks. Compared to traditional classifiers, which divide pixels based on specific spectral information, FCNs use multiple fully convolutional layers to extract embedded high-order context features in images, achieving pixel-level annotations.

Pixel-based land use/cover sample sets are similar to remote sensing semantic segmentation datasets, with the labeling process mainly involving the annotation of all pixels that cover a specific land object (as shown in Figure 8). The advantage for this type of sample set is that it can obtain accurate boundaries for land objects, but the downside is that the labeling workload is relatively large. Common pixel-level sample sets are shown in Table 4. From this table, it can be seen that most sample sets are limited by spectral resolution, resulting in lower spatial resolutions for these datasets. Mostly, each dataset only contains a single image and annotations for a specific study area, with only a few recently published datasets (such as DeepGlobe [69]) having higher numbers of samples and spatial resolutions. However, these datasets only consist of ordinary RGB or RGBNIR images with lower spectral resolutions.

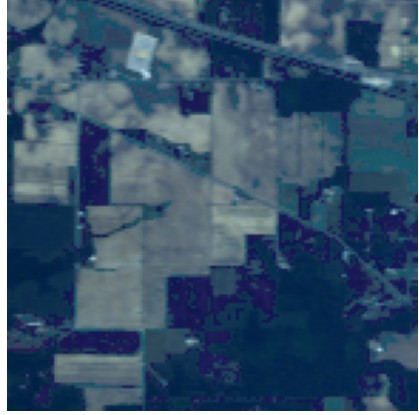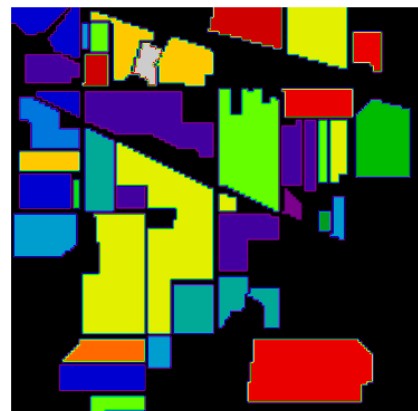

**Figure 8.** Indian pines example sample data [66].

**Table 4.** Pixel-level LULC dataset.

| Name | Source | Size | Resolution | Dimensions | Classes |
|---|---|---|---|---|---|
| Indian Pines [66] | AVIRIS | 145 × 145 | 20 m | 224 | 16 |
| DeepGlobe [69] | Mix | 2448 × 2448 | 0.5 m | 3 | 7 |
| Salinas [66] | AVIRIS | 521 × 127 | 3.7 m | 224 | 16 |
| University of Pavia [66] | ROSIS | 610 × 610 | 1.3 m | 103 | 9 |

4.1.2. Object-Level LULC Remote Sensing Classification Dataset

Scene-based classification, also known as patch-based classification, primarily divides images into a series of LULC categories based on the main content of the images [70]. Typical scene-based classification methods will first sample a large number of patches from a larger image for model training. Then, by classifying the scene category of each sampled patch, a trained model will generate the LULC map [71]. However, scene-based CNNs have some limitations in LULC classification. First, it is challenging to define suitable patch sizes, especially when there is significant variation in the sizes of the ground targets [72]. Additionally, sampled patches within the same large image will be processed independently, which means contextual information is ignored during the classification process. Scene-based methods are usually used to identify large objects, while pixel-based methods are more suitable for detecting subtle details [73].

The image-block-based LULC sample dataset was similar to the remote sensing object recognition dataset, and its labeling process mainly involved assigning specific LULC categories to individual N × N image blocks (as shown in Figure 9). DL models that correspond to this type of dataset are typically based on CNN or RNN image classification models. The advantage of this approach is the simplicity of the labeling process, but the drawback is the inability to obtain boundary information for specific objects. Table 5 lists widely used and influential image-block-level sample datasets, along with relevant information about this data.

**Table 5.** Object-level LULC dataset.

| Name | Source | Size | Dimensions | Resolution | Classes |
|---|---|---|---|---|---|
| NWPU-RESISC45 [70] | Google Earth | 256 × 256 | 0.2∼30 m | 3 | 45 |
| UC Merced Dataset [57] | Aerial Images | 256 × 256 | 0.3 m | 3 | 21 |
| EuroSAT [64] | Sentinel-2 | 64 × 64 | 10/20/60 m | 13 | 10 |
| WHU-RS19 [58] | Google Earth | 600 × 600 | 0.5 m | 3 | 19 |
| RSSCN7 [59] | Google Earth | 400 × 400 | 0.25∼2 m | 3 | 7 |

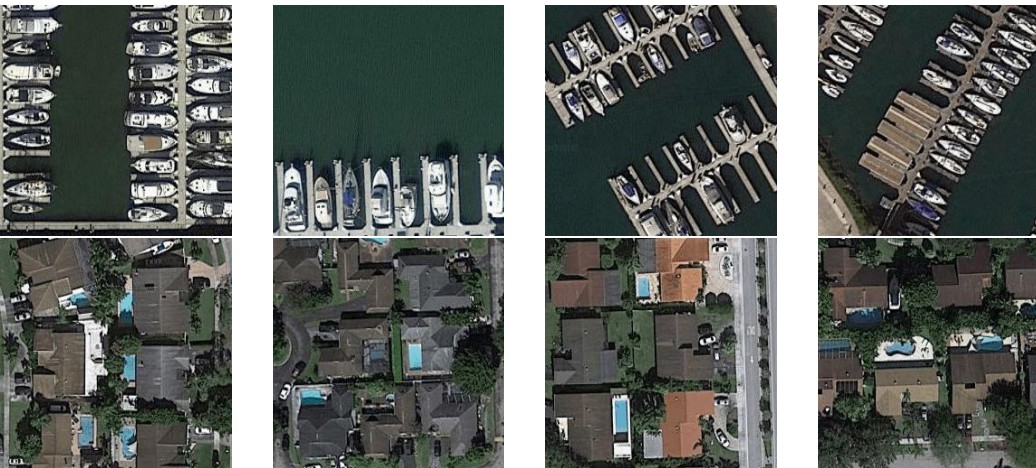

**Figure 9.** NWPU-RESISC45 example sample data [70].

*4.2. Deep Learning Model*

The design of using DL models for LULC research can be divided into the following four steps [74]: (1) data preprocessing: performing denoising, fusion, dimension reduction, resampling, and other processes on image data; (2) model training: after preprocessing of image data, using these data to train DL models; (3) validation and evaluation: assessing the accuracy of the trained models to ensure their performance; (4) LULC mapping: predicting land use and land cover maps to assist urban planners and land resource managers in making appropriate decisions Figure 10.

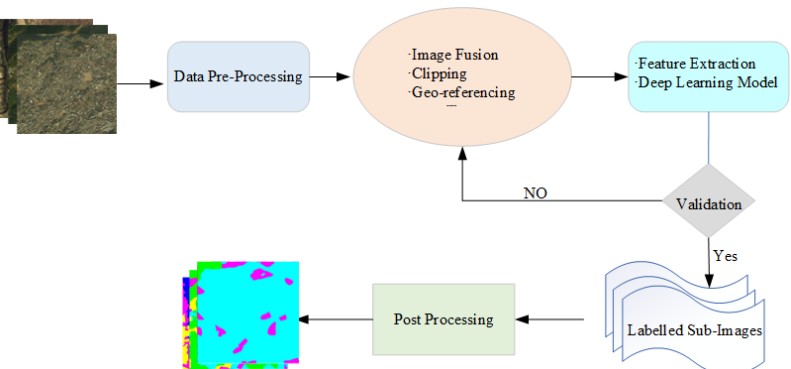

**Figure 10.** An overall framework of a DL model for LULC.

CNNs are one of the most commonly used DL models in LULC [75]. In the following section, we delve into the prevalent network architectures employed in the classification of LULC in remote sensing. These predominantly include CNNs, fully convolutional networks models (FCNs) and recurrent neural network (RNNs).

4.2.1. Convolutional Neural Networks Model

CNNs are a DL approach that has achieved remarkable success in tasks such as image classification, object detection, and semantic segmentation. They are widely used in image classification and computer vision tasks [76]. Due to the powerful feature extraction capability of CNNs, numerous studies have been conducted on LULC classification based on them, and significant achievements have been made therein [77–80] (Figure 11 illustrates the workflow of using CNNs for LULC). Early studies concerning LULC based on DL mostly focused on feature representation or learning, while the final classification used other simpler classifiers [81]. Verma D. et al. proposed a novel solution to generate classification maps with a 10-band Sentinel-2B dataset and a CNN at a 10 m spatial resolution [77]. Marcos et al. [82] developed a convolutional neural network (CNN) architecture with

rotation equivariance, applying it to two sub-decimeter land cover semantic labeling benchmarks for a more accurate mapping of LULC. To prevent overfitting during the model training process, Zhu et al. [83] introduced generative adversarial networks (GAN) as a regularization technique into the CNN model for hyperspectral remote sensing image classification. These methods have demonstrated that CNNs have made a profound and meaningful impact in the field of LULC.

The architecture of CNNs is an ensemble of convolutional layers, max pooling layer, and fully-connected layers [84]. In a convolutional layer, a filter is applied on the preceding feature, producing a weighted sum that passes through an activation function to yield the final result. This methodology calculates the kernel size to find local correlations and maintain invariance within the data array. The outcome is a feature map that exhibits invariance to the smallest possible unit. Subsequently, a fully-connected neural network integrates different convolution or pooling layer phases together [85].

A convolution operation can be described as:

$$f_{a,b}(x,y) = \sum_i \sum_d h_{i,j}(s,t)c_d \tag{1}$$

$$F_{i,j} = [f_{a,b}(1,1), \ldots, f_{a,b}(x,y), \ldots, f_{a,b}(X,Y)] \tag{2}$$

Following feature extraction, pooling or down-sampling operation comes into play which forms a blend of features that are robust to minor distortions and translational shifts.

$$P_{i,j} = g_p(F_{i,j}) \tag{3}$$

Here, $P_{i,j}$ is the pooling feature-map of the $i$-th layer for the $j$-th input feature-map and $g_p$ symbolizes the pooling operation. Various pooling operations are utilized in CNN such as max, average, L2, overlapping, and spatial pyramid pooling. Activation functions speed-up the learning and provide a decision function for a convolved feature-map.

$$t_{i,j} = g_a(F_{i,j}) \tag{4}$$

In the above equation, $g_a$ denotes the activation function and $F_{i,j}$ denotes the convolution output, $t_{i,j}$ signifies the transformed output.

The training and optimization of CNN are crucial design aspects that ensure optimum performance and manage overfitting. As data volume surges, the number of potential challenges during the training process also amplifies. Overfitting can be controlled by strategies such as dropout and batch normalization. Dropout deactivates multiple nodes at the conclusion of each training cycle. Batch normalization aims to enforce a zero mean and a one standard deviation for all activation functions in the specified layer for each small batch, enhancing overall accuracy, making the network resilient to overfitting and expediting the convergence of the gradient descent process. The last part of the CNN model is the fully-connected layer that interlinks each layer with another one to classify, it conducts an analysis on the output of all previous levels and classifies data by connecting selected features non-linearly [86].

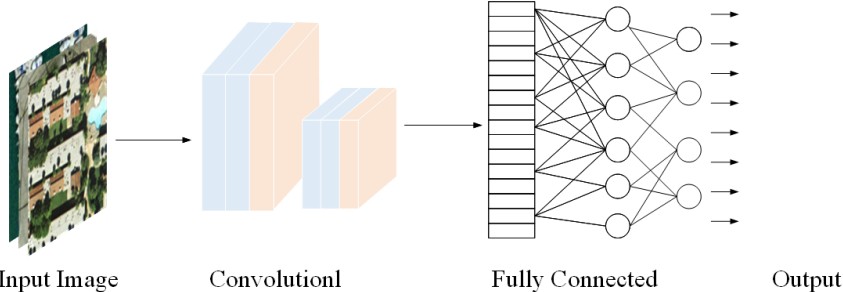

**Figure 11.** Convolutional neural networks model.

### 4.2.2. Fully Convolutional Networks Model

Semantic segmentation with FCNs is a pivotal DL technique for LULC applications, assigning land cover labels to each pixel in an image [87]. The FCNs structure, consisting of an encoder-decoder framework, provides an end-to-end approach, especially useful for remote sensing imagery analysis (The workflow of using FCNs for LULC is depicted in Figure 12). The encoder, responsible for feature extraction, applies a convolution operation, mathematically represented as:

$$(I * K)(i, j) = \sum_m \sum_n I(i - m, j - n)K(m, n) \tag{5}$$

where $I$ is the input, $K$ is the kernel, and $(i, j)$ are the spatial dimensions. To maintain the input's height and width, zero-padding is utilized. Following convolution, the pooling function reduces the dimensionality of the input image, symbolically represented as:

$$P(i, j) = \max_{(x,y) \in N(i,j)} I(x, y) \tag{6}$$

where $P$ is the pooled output, $N(i, j)$ denotes the neighborhood of pixel $(i, j)$, and $I(x, y)$ is the intensity of pixel at $(x, y)$. This step eliminates less important features while retaining essential ones.

The encoder's output and the upsampled decoder output are concatenated, doubling the height and width of the image in the up-sampling layer. Lastly, the deconvolution operation, the inverse of the convolution, generates the final output.

Wurm Michael et al. [79] proposed a fast fully convolutional network (FastFCN) to semantically segment satellite images and thus classify LULC. Alhassan V. et al. proposed an FCN that would incorporate a context module and an adversarial extension was proposed to enhance the quality of generated LULC maps [80]. Sertel E. et al. [88] proposed a model based on the DeepLabv3+ architecture with a ResNeXt50 encoder for the semantic segmentation of very high-resolution (VHR) Worldview-3 satellite images, subsequently facilitating LULC classification. Balancing context information extraction and accurate boundary localization remains a task, due to the strong downsampling and local detail requirements. Despite this, the utility of FCNs spans various applications within remote sensing LULC.

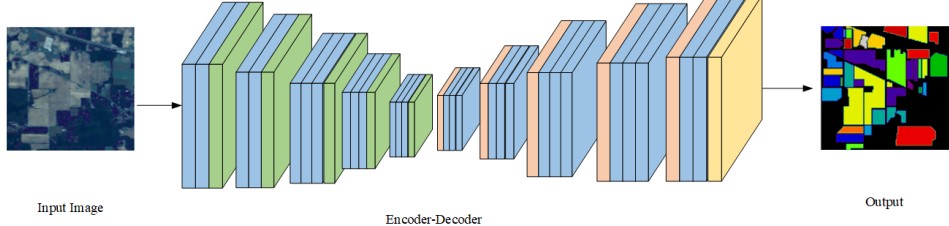

**Figure 12.** Fully convolutional networks model.

### 4.2.3. Recurrent Neural Network, Long Short Term Memory Network and Gated Recurrent Unit Model

Real-world datasets often encapsulate sequential data with inherent correlations, posing a challenge to conventional network models due to their inefficiency in handling such correlations [89]. RNNs, LSTMs, and GRUs demonstrate proficiency in this domain, leveraging these inherent correlations for processing and generating sequences [90]. An RNN handles sequential data by utilizing hidden states; the activation of each hidden state depends on the previous stages. However, traditional RNNs will encounter the issue of vanishing or exploding gradients when dealing with long-term sequential data [91]. Therefore, this problem is addressed by introducing LSTM and GRU architectures [92,93] (The flowchart of the LSTM network is illustrated in Figure 13). Lyu H. et al. [94] proposed an RNN for LULC change detection as they realized that RNNs have an advantage in

solving challenging problems that involve sequential time series data. This significant and meaningful work tackled long-time-series data analysis issues. Jeyavathana et al. [95] proposed the use of LSTM in RNNs to achieve high levels of classification accuracy and to solve the potential memory issues of internal states. Luo D. et al. [96] proposed using the CNN-LSTM model to classify changing land use and land cover (LULC) over time in the agricultural expansion area of the Matopiba region in Brazil. Comparisons were made with other methods such as CNN and CNN-GRU, demonstrating their reliability for both coarse and medium spatial resolution satellite images.

In an RNN, let us denote the input sequence as $\mathbf{X} = (X_1, \ldots, X_t, X_{t+1})$, the hidden state as $\mathbf{H} = (H_1, \ldots, H_t, H_{t+1})$, and the output as $\mathbf{Y} = (Y_1, \ldots, Y_t, Y_{t+1})$. These elements of the input sequence are sequentially ingested by the RNNs, yielding corresponding output sequence units for each phase and information to be utilized in the subsequent phase, thus capitalizing on the correlation between sequences.

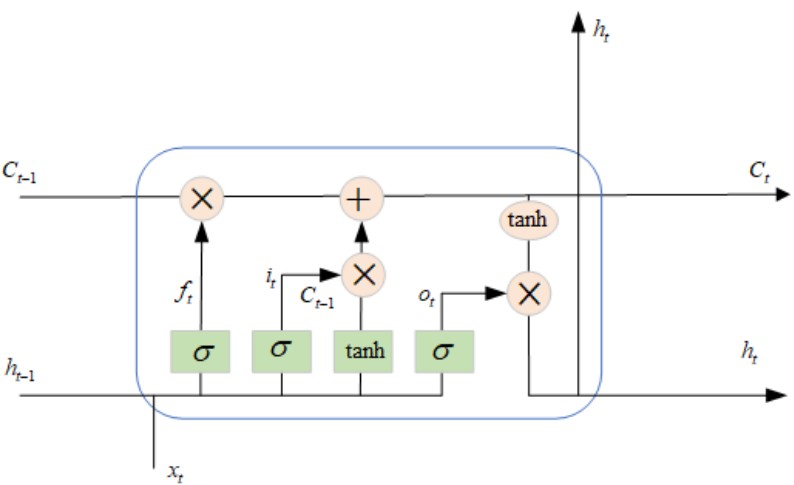

**Figure 13.** Long short term memory network model.

4.2.4. Autoencoder Model

An autoencoder (AE) is a central methodology in deep learning, crafted for hierarchical feature representation. The AE's architecture consists of three main layers: an input layer (also known as the encoding layer), a hidden layer, and an output layer (also referred to as the reconstruction or decoding layer). The hidden layer typically contains fewer nodes compared to both the input and output layers, which have an identical count of nodes. Each pair of layers employs a non-linearity function. The AE transforms an input layer denoted by $p$ from $P^n$ to a hidden layer denoted by $q$ from $Q^h$, creating a latent representation. Here, $Z$ represents the weight matrix of the input, $\alpha$ denotes the bias vector for the hidden layer, and $f()$ signifies the activation function. Thus, we have:

$$q = f(Zp + \alpha) \tag{7}$$

After this step, the latent vector $q$ is utilized to reverse map to output $t$, belonging to $T^n$, where

$$t = f(\lambda q + \delta) \tag{8}$$

In the equation above, $t$ stands for the output layer, $\lambda$ represents the weight matrix transitioning from the hidden layer to the output layer, and $\delta$ is the output layer's bias vector. The objective during training is to minimize the reconstruction error denoted by $e(p, t)$ between $p$ and $t$.

$$e(p, t) \tag{9}$$

If this error is below a certain threshold, the latent representation becomes useful for minimizing feature count. An illustration of the AE structure is depicted in Figure 14.

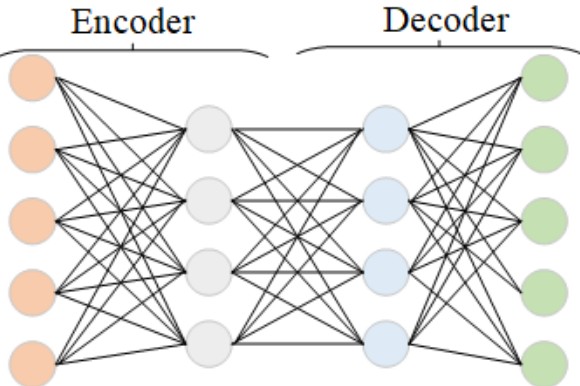

**Figure 14.** Autoencoder model.

4.2.5. Adversarial Extension Model

Generative adversarial networks (GANs) are a type of deep learning model, designed for synthesizing data, with applications in Remote Sensing (RS) including LULC, classification, and super-resolution. The process of Generative Adversarial Networks (GANs) 479 is depicted in Figure 15. Singh A et al. [97] introduced a data augmentation technique based on spectral index generative adversarial networks to train deep convolutional neural networks, utilizing the spectral features of multispectral images to support data augmentation, aimed to mitigate the issue of limited LULC sample data. He C et al. [98] proposed an end-to-end GAN integrated with a conditional random field for the semantic segmentation of RS images. The integration of the skip-connected encoder-decoder generator with the CRF layer aids in extracting better local and global information from the images. The essence of GANs is two interactive models: a generator ($g$) and a discriminator ($d$). The generator's role is to fabricate data, and the discriminator's is to differentiate between real and synthetic data.

In the framework of GANs, we define a dataset $M$ comprising of $m$ training images and corresponding ground-truth maps. The generator is a conditional probability model, trained to produce maps akin to ground truth. Conversely, the discriminator, based on a joint probability model, aims to correctly identify the ground truth maps and discriminate between these and maps produced by the generator.

The generator loss function integrates the multi-class entropy loss ($L_{mce}$) with a binary class entropy loss ($L_{bce}$). The multi-class entropy loss is defined as:

$$L_{mce}(a, \hat{a}) = -\sum_{i=1}^{I} a_i \log(\hat{a}_i), \tag{10}$$

and the binary class entropy loss as:

$$L_{bce}(b, \hat{b}) = -(b \log(\hat{b}) + (1 - b) \log(1 - \hat{b})), \tag{11}$$

where $a$ is ground-truth, $\hat{a}$ is the predicted output, $I$ is the number of classes, and $b$ is the binary probability for predicted output and ground truth, while $\hat{b}$ is the predicted probability between 0 and 1.

The discriminator aims to minimize the loss function:

$$\sum_{j=1}^{J} L_{bce}(d(x_j, y_j), 1) + L_{bce}(d(x_j, g(x_j)), 0), \tag{12}$$

while the generator minimizes the multi-class entropy loss but also tries to degrade the performance of the discriminator. This results in the following loss function:

$$\sum_{j=1}^{J} L_{mce}(g(x_j), y_j) + \lambda L_{bce}(d(x_j, g(x_j)), 1), \tag{13}$$

where $\lambda$ is a regularization constant.

The parameters of the generator and discriminator, represented by $\theta_g$ and $\theta_d$, respectively, are optimized by minimizing a hybrid loss function:

$$L(\theta_g, \theta_d) = \sum_{j=1}^{J} L_{mce}(g(x_j), y_j) - \lambda(L_{bce}(d(x_j, y_j), 1) + L_{bce}(d(x_j, g(x_j)), 0)). \tag{14}$$

In this adversarial setting, pre-trained base networks are fine-tuned with the discriminator to maximize the overall performance.

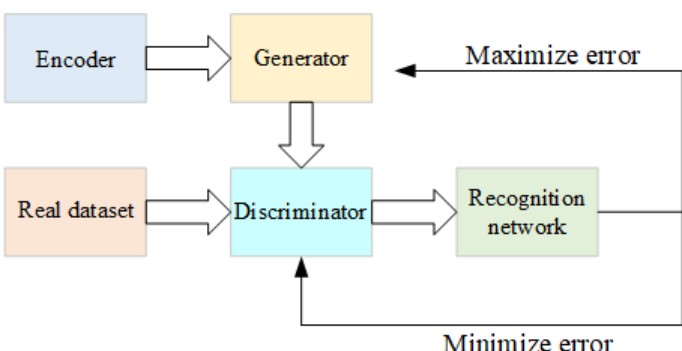

**Figure 15.** Generative adversarial networks model.

In the analysis of literature data spanning 2013 to 2023, we identified patterns in the usage of different DL models for LULC classification tasks. The corresponding bar chart in Figure 16 captures the frequency of model usage over these years. CNNs emerged as the most commonly employed model throughout the given period. This trend can be attributed to CNN's proficiency in image classification tasks, which is a critical aspect of LULC analysis. FCNs, a variant of CNNs tailored for semantic segmentation tasks, also demonstrated significant representation. Their ability to handle spatially organized image data makes them well-suited for LULC classification. RNNs and their variants, LSTM and GRU, were used less frequently. This is likely due to their optimal applicability to time-series data rather than spatial data, although their usage in combination with CNNs for spatio-temporal data was noted. In summary, the literature data reveals a growing and diverse application of DL models in LULC tasks over the past decade.

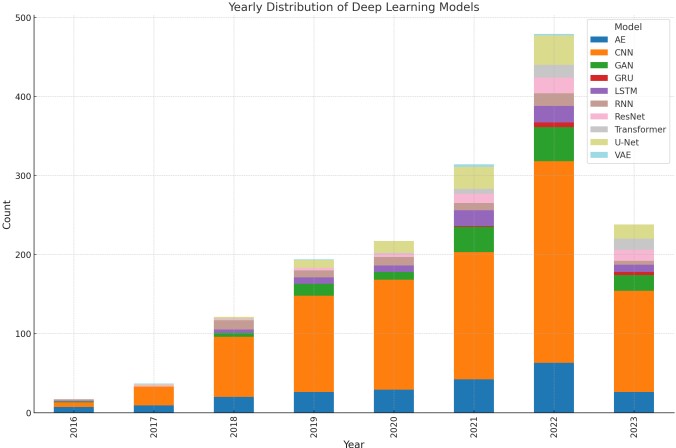

**Figure 16.** Distribution of DL models used in LULC from 2013 to 2023.

## 5. Conclusions

This study employed topic-driven bibliometric and knowledge graph analysis methods to examine academic articles about the application of DL technology in LULC research from 2013 to 2023. These research findings not only highlighted the main research questions of interest for scholars but also identified the article and citation counts, primary sources of publications, themes, institutions, and countries/regions of origin, as well as visualizing the academic collaboration relationships.

This study contributes to the LULC field in three ways. First, it is intended to aid researchers and policy-makers in this field to better comprehend the history of, present status of, and future trends of in DL technology in LULC research. Second, with its study of high-yield institutions and countries/regions, it can help scholars, particularly newcomers to the field, share research findings, collaborate, and identify key participants in the application of DL technology in the LULC field. Third, the results of this analysis of high-yield publication sources and themes can provide scholars with guidance regarding where to submit their work. Most importantly, word cloud and theme analysis can offer researchers insights into the important research issues in this field.

**Author Contributions:** Conceptualization, C.R. and W.F.; methodology, C.R.; validation, C.R. and W.F.; formal analysis, C.R. and W.F.; investigation, C.R.; resources, W.F.; data curation, W.F.; writing—original draft preparation, C.R. and W.F.; writing—review and editing, W.F.; visualization, C.R.; supervision, W.F.; project administration, W.F.; funding acquisition, W.F. All authors have read and agreed to the published version of the manuscript.

**Funding:** This study was supported by the National Natural Science Foundation of China (Project No. 61971417).

**Data Availability Statement:** The data presented in this study are available on request from the corresponding author. The data are not publicly available due to privacy restrictions.

**Conflicts of Interest:** The authors declare no conflict of interest.

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
