# Peer review of "A Comprehensive Review of Land Use and Land Cover Change Based on Knowledge Graph and Bibliometric Analyses"

_land, doi:10.3390/land12081573_

Round 1

Reviewer 1 Report

Dear authors,

The manuscript titled: “A Comprehensive Review of Land Use and Land Cover Change Studies Using Deep Learning: Insights from Knowledge Graphs and Bibliometric Analysis” was evaluated. The assessment was carried out very carefully.

Main points to review:

1. The introductory part can be improved and in this sense gain greater impact: authors should take advantage of the great impact of land cover and land use studies in the world and expand the introduction;

Studies on land cover and use stand out from the perspective of impacts caused by climate change as well as anthropic activities over time. Mainly in semi-arid regions. The authors could improve this paragraph in this sense or detail this amplitude in another;

Reinforcing this subject with bibliographical references of LULC from different parts of the world is also important.

2. The authors could present more methodological details, a kind of step-by-step, important for the replication of this type of study in the future.

Author Response

Point 1: Enhance the introduction to stress the global significance of LULC studies, underlining their role in climate change and human activities, supplemented with a diverse range of bibliographic references.

Response 1: We greatly appreciate your suggestions. In response, we have enriched our paper by highlighting the significance of LULC in relation to climate and human activities.

Point 2: Provide more methodological details in a step-by-step format, which will be beneficial for future replication of this kind of study.

Response 2: Thank you for your suggestion. We have revised the methodology section of our paper, providing a more detailed description of the methods used in our research.

(The revisions based on your suggestions have been highlighted in yellow in the modified PDF.)

Reviewer 2 Report

Dear Authors,

The article offers a solution to application review of land use and land cover change studies using deep learning based on the Bibliometric analysis. For this purpose, a land-use change and deep learning was used through bibliometric analysis. I really enjoyed reading and revising it.

Datasets used are clearly defined and analysis methods are sufficiently explained and justified. Results are straightforward and conclusions are properly discussed. The content and theme of the article is consistent with the lines of the journal and the topic is of interest to the readers.

The paper contain all main obligatory chapters (Introduction; Materials and Methods; Results; Discussion). However, the paper has some inconsistent compering to the instructors for the authors, which should be corrected. My corrections are the following ones:

TITLE

The title of the manuscript are concise, specific and relevant. This is Ok.

ABSTRACT

The Abstract contain all main obligatory elements. But  It exceeds 200 words. According to the instruction to the authors you can revised.

List of a Keywords is appropriate. But word of “Knowledge Graphkeyword” is not in abstract. It can be change acording to discretion of authors.

INTRODUCTION

Land use land cover (LULC) and land use change can be defined based on the literature in the first paragraph (20-30)

Deep learning models for LULC monitoring are discussed in the introduction. İt is ok. Introduction chapter do contain all mandatory elements except the specific hypothesis.

The most cited publications in the field of land use (Wos) can be discussed.

MATERIALS AND METHODS

Acronyms/Abbreviations/Initialisms are defined when they first time appear in each of three sections: the abstract; the main text; the first figure or table. However, suggestion is that abbreviation or its explanation are not in chapters titles. For exaple Web of Science (WOS) (84)

It was stated that bibliometrix and citespace were used in graphic analysis. it is seen that VOSviewer is used in some graphics. It should also be written that VOSviewer is used in graphic analysis.

RESULT

Figure 1 not readable. According to journal rule authors can be revised.

Figure 6 legend can be revised.

Some graphics and tables are not cited in the paper. For exaple fugure 3,4.

Keyword analysis can be added as words depending on the authors' decision.

The Result chapter provide concise and precise description of the experiment results.

DISCUSSION

This chapter discusses deep learning methods instead of discussing the bibliometric analyzes mentioned earlier.

CONCLUSION

This chapter is integrated with discussion chapter and present results of the research. This is Ok.

REFERENCES

The references are numbered in order of appearance in the text in the text in square bracket. This is Ok.

All equations are numbered in brackets and placed on the right margin of the text. This is Ok.

Text font in tables and figures should be according to the instructions for the authors.

Author Response

Response to Reviewer 2 Comments

Point 1: I notice that the abstract in your manuscript exceeds the recommended length of 200 words. It would be beneficial if you revise the abstract to adhere to the guidelines.

Response 1: Thank you for your suggestion about the abstract's length. We have meticulously reviewed and revised this section to ensure conciseness while maintaining its comprehensiveness. The abstract now aligns with the journal's guidelines of having 200 words or less. We appreciate your attention to this detail, which indeed enhances the readability of the manuscript

Point 2: In your list of keywords, you included the term "Knowledge Graphkeyword", however, this term is not mentioned in your abstract. It would be appropriate to revise either your keyword list or your abstract to maintain consistency.

Response 2: We appreciate your meticulous review of our paper. We have made adjustments concerning the keyword issue as suggested.

Point 3: The introduction section needs to be revised to include a definition of LULC and to elaborate on the impacts of LULC.

Response 3: Thank you for your valuable suggestion. We have expanded the profound impacts of LULC in the introduction and have also included its definition.

Point 4: Acronyms/Abbreviations/Initialisms are defined when they first time appear in each of three sections: the abstract; the main text; the first figure or table. However, suggestion is that abbreviation or its explanation are not in chapters titles. For exaple Web of Science (WOS) (84)

Response 4: Thank you for your thorough reading of our paper. Based on your suggestions, we have meticulously reviewed our work word by word and corrected the errors.

Point 5: It was stated that bibliometrix and citespace were used in graphic analysis. it is seen that VOSviewer is used in some graphics. It should also be written that VOSviewer is used in graphic analysis.

Response 5: Thank you for your thorough reading of our paper. Based on your suggestions, we have rectified the errors in our paper.

Point 6: Figure 1 not readable. According to journal rule authors can be revised.Figure 6 legend can be revised.Some graphics and tables are not cited in the paper. For exaple fugure 3,4.

Response 6: Thank you. We have improved the quality of Figure 1 and revised the legend of Figure 6. Furthermore, we have checked and clearly indicated all the references to figures and tables in the paper.

Point 7: Figure 1 not readable. According to journal rule authors can be revised.Figure 6 legend can be revised.Some graphics and tables are not cited in the paper. For exaple fugure 3,4.

Response 7: Thank you. We have improved the quality of Figure 1 and revised the legend of Figure 6. Furthermore, we have checked and clearly indicated all the references to figures and tables in the paper.

Point 8: This chapter discusses deep learning methods instead of discussing the bibliometric analyzes mentioned earlier.

Response 8: Thank you for your suggestions. The narrative flow of our paper is designed in the following manner: initially, we use bibliometric analysis and knowledge graph analysis to observe the research trends and hotspots in this field over the past decade. In the discussion section, we aim to discuss the existing problems and possible future research directions in this field. Therefore, in the discussion section, we primarily describe the current state of deep learning applications in LULC.

(The revisions made based on your suggestions have been highlighted in green in the modified PDF.)

Reviewer 3 Report

I have checked the paper but the major revision need for publication, Before publication author must improve language of the paper. Detailed comments attached in the PDF file.

Author Response

Response to Reviewer 3 Comments

Point 1: Expand the introduction section.

Response 1: Thank you for your suggestion. We have revised the introduction according to your recommendation.

Point 2: Author should refine the entire paper improve the English language.

Response 2: Thank you for your careful reading of our paper. We have enlisted the services of MDPI English editing to make language modifications to the article.

Point 3: Improved the quality of figures1.

Response 3: Thank you for your suggestion, we have made modifications to Figure 1 accordingly.

Point 4:Separate discussion section must add in the paper.

Response 4: Thank you for your careful reading of our manuscript. I must apologize for my oversight. In the process of writing with LaTeX, I did not notice my mistake in the commands, including during later reviews. I have now made the necessary corrections.

Point 5: Author must add the another Model information.. Missing so many important model.

Response 5: Thank you for your suggestion. Originally, this section was designed to introduce some simple and common models used in the LULC field. Now, in response to your advice, we have added some novel models, including the likes of AE and GCN.

Point 6: How to find out the Figure16 detailed source add in the paper. Author add the research gap, Feature road map,  and Limitation etc. must add in the paper.

Response 6:Thank you for your feedback. We used Python to identify the models used in articles from the 'abstract' field of the exported bibliographic data, then compiled the volume of data using these models in the papers published each year.

Point 7: Author told 1342 paper Reviewed but the In the paper only  55 papers. So many database missing in the paper. at least above 100 to 200 important reference need in the papers. I think author does not review s the entire documents.

Response 7: Thank you for your careful reading of our manuscript. The 1342 papers mentioned in the text represent the total number of publications on the application of deep learning in the field of LULC over the past decade. We utilized this total volume to examine the research trends, leading countries and regions in publication volume, and the main research themes, among others. However, the manuscript does not cite content from all 1342 papers; 55 references were used as some phrases in our text were indeed quoted from these papers. We have now supplemented the manuscript based on your suggestions and added some references. Here are a few examples of similar articles we referred to for the treatment of reference citations: For instance, based on 2572 publications related to AI-supported brain magnetic resonance imaging processing, Chen et al. provided a topic-driven bibliometrics to examine the status, tendencies, research topics, and promising directions concerning this area, yet their paper only cites 146 references(Chen X, Zhang X, Xie H, et al. A bibliometric and visual analysis of artificial intelligence technologies-enhanced brain MRI research[J]. Multimedia Tools and Applications, 2021, 80: 17335-17363.); Xieling Chen and colleagues used 386 papers to perform a bibliometric analysis on the development of knowledge graph over nearly thirty years, and their paper cites 115 references(Chen X, Xie H, Li Z, et al. Topic analysis and development in knowledge graph research: A bibliometric review on three decades[J]. Neurocomputing, 2021, 461: 497-515.).

(The revisions made based on your suggestions have been highlighted in pink in the modified PDF.)

Round 2

Reviewer 1 Report

Dear authors,

In view of all the corrections made in a significant way, I consider this manuscript as approved in this version.

Reviewer 3 Report

no more comments